# *Mycobacterium tuberculosis* Fatty Acyl-CoA Synthetase *fad*D33 Promotes *Bacillus Calmette–Guérin* Survival in Hostile Extracellular and Intracellular Microenvironments in the Host

**DOI:** 10.3390/cells12222610

**Published:** 2023-11-11

**Authors:** Yifan Zhu, Hongling Shi, Tian Tang, Qianqian Li, Yongchong Peng, Luiz E. Bermudez, Changmin Hu, Huanchun Chen, Aizhen Guo, Yingyu Chen

**Affiliations:** 1State Key Laboratory of Agricultural Microbiology, College of Veterinary Medicine, Huazhong Agricultural University, Wuhan 430070, China; zhuyifan@webmail.hzau.edu.cn (Y.Z.); conky@webmail.hzau.edu.cn (T.T.);; 2National Animal Tuberculosis Para-Reference Laboratory (Wuhan) of Ministry of Agriculture and Rural Affairs, International Research Center for Animal Disease, Ministry of Science and Technology, Huazhong Agricultural University, Wuhan 430070, China; 3Department of Biomedical Sciences, College of Veterinary Medicine, Oregon State University, Corvallis, OR 97331, USA

**Keywords:** *fad*D*33* mutant, extracellular survival, intracellular survival, p38MAPK, inflammatory cytokines

## Abstract

Tuberculosis, caused by *Mycobacterium tuberculosis* (*M. tb*), remains a significant global health challenge. The survival of *M. tb* in hostile extracellular and intracellular microenvironments is crucial for its pathogenicity. In this study, we discovered a *Bacillus Calmette–Guérin* (BCG) mutant B1033 that potentially affected mycobacterium pathogenicity. This mutant contained an insertion mutation gene, *fad*D33, which is involved in lipid metabolism; however, its direct role in regulating *M. tb* infection is not well understood. Here, we found that the absence of *fad*D*33* reduced BCG adhesion and invasion into human pulmonary alveolar epithelial cells and increased the permeability of the mycobacterial cell wall, allowing *M. tb* to survive in the low pH and membrane pressure extracellular microenvironment of the host cells. The absence of *fad*D*33* also inhibited the survival of BCG in macrophages by promoting the release of proinflammatory cytokines, such as interleukin (IL)-1β, IL-6, and tumors necrosis factor-α, through the mitogen-activated protein kinase p38 signaling pathway. Overall, these findings provide new insights into *M. tb* mechanisms to evade host defenses and might contribute to identifying potential therapeutic and vaccine targets for tuberculosis prevention.

## 1. Introduction

Tuberculosis, primarily caused by *Mycobacterium tuberculosis (M. tb*), was the leading cause of death from a single infectious agent until the emergence of the coronavirus according to the World Health Organization. In 2021, approximately 6.4 million new cases of tuberculosis and 1.4 million tuberculosis-related deaths were reported among human immunodeficiency virus-negative people worldwide [1].

*M. tb* is an intracellular bacterium that has coevolved with humans due to its obligatory intra-macrophage phagosome habitat and slow replication ability. In addition, *M. tb* uniquely avoids innate immunity in a highly complex manner [2]. *M. tb* encounters various hostile environments within the host, such as low oxygen (hypoxia), nutrient scarcity, acidic pH, and membrane tension [3,4,5]. Consequently, *M. tb* deploys strategies to survive in cells and appropriately respond to the microenvironment [6,7]. These strategies involve preventing the fusion and acidification of the phagolysosome, suppressing phagosome maturation and autophagy, influencing macrophage apoptosis and granuloma formation, and modulating immune responses [6,8].

Previous studies have reported that *M. tb* has 36 *fad*D genes, which are involved in both lipid biosynthesis and catabolism [9]. *fad*D13, a peripheral membrane-associated acyl-CoA ligase, activates C26 and C24 fatty acids for the *M. tb* mymA operon, which synthesizes cell envelope under acidic conditions [10,11]. The acyl-CoA ligase *fad*D5, located within the mce1 operon of *M. tb*, is involved in the growth of *M. tb* using mycolic acids as the sole carbon source. *fad*D3 functions as an acyl-CoA ligase in cholesterol catabolism [12]. *fad*D26 is involved in the biosynthesis of phthiocerol and phthiodiolone dimycocerosate, whereas *fad*D22 and *fad*D29 are involved in the biosynthesis of phenolic glycolipids [13]. *fad*D*33*, a fatty acyl AMP ligase, is involved in the synthesis of mycobactin [14] and participates in bacterial virulence by promoting bacterial proliferation in the liver [15]; however, its role in *M. tb* infection remains unknown.

In the present study, we screened a *fad*D*33* mutant using transposon mutant libraries constructed in our laboratory. We found that the absence of *fad*D*33* in *Bacillus Calmette–Guérin* (BCG) reduced adhesion and invasion abilities, increased colony size, increased mycobacterial cell wall permeability, and decreased BCG survival in low pH and stress conditions. Moreover, the absence of *fad*D33 in BCG inhibited the intracellular survival of bacteria by activating the p38 MAPK signaling pathway and promoting the synthesis of inflammatory cytokines (interleukin [IL]-1β, IL-6, and tumors necrosis factor-α [TNF-α]). Our study suggests that *fad*D*33* is a potential therapeutic and vaccine target for tuberculosis prevention.

## 2. Materials and Methods

### 2.1. Bacterial Strains, Cells, and Culture Conditions

BCG was cultured in Middlebrook 7H9 medium (BD PharMingen, San Diego, CA, USA) with 10% oleic acid, albumin, dextrose, and catalase (OADC, BD PharMingen); 0.05% Tween 80 (Sigma-Aldrich, St. Louis, MO, USA); and 0.2% glycerol (Sigma-Aldrich) or on MB 7H11 agar plates (BD PharMingen) containing 0.5% glycerol and 10% OADC.

DH5α and BL21 (DE3) *E. coli* strains were grown in Luria broth under shaker conditions at 180 rpm. Antibiotics were added at the following concentrations: kanamycin: 50 μg/mL and hygromycin: 150 μg/mL. All cultures were incubated at 37 °C.

A549 cells were cultured in Dulbecco’s modified Eagle’s medium (DMEM, HyClone, Logan, UT, USA) supplemented with 10% fetal bovine serum (FBS, Gibco, Grand Island, NY, USA). THP-1 cells were maintained in RPMI medium (HyClone) supplemented with 10% FBS. Cells were cultured at 37 °C in a humidified atmosphere with 5% CO_2_.

### 2.2. Construction of a Transposon Library

A BCG transposon library was constructed using Himar1 mutagenesis. In brief, mid-log phase BCG cultures were incubated with ΦMycoMarT7 phages at 37 °C for 10–18 h. Afterward, cultures were washed, resuspended in 25% glycerol, plated onto 7H11 medium with kanamycin, and incubated at 37 °C for 21 days.

One hundred colonies were randomly selected and screened for the presence of the kanamycin gene via polymerase chain reaction (PCR) using the primers 5′-AAGATGGATTGCACGCAGGT-3′ (forward) and 5′-AAGAACTCGTCAAGAAGG-CGA-3′ (reverse). PCR cycling conditions were as follows: 95 °C for 5 min, followed by 35 cycles of 95 °C for 45 s, 55 °C for 1 min, and 72 °C for 1 min. Then, the reactions were maintained at 72 °C for 10 min. The colonies that were kanamycin resistant were grown in 7H9 broth containing kanamycin. A library of 3034 individual transposon mutants was generated.

### 2.3. Invasion, Adhesion, and Infection Assay

A549 cells were infected with live BCG at a multiplicity of infection (MOI) of 10 for 1 h at 37 °C for invasion assay and 30 min at 4 °C for adhesion assay. Then, the cells were washed three times with sterile 1× phosphate-buffered saline (PBS, Gibco) and incubated with 1 mL DMEM containing 10% FBS. For the invasion assay, 100 μg/mL gentamicin was also added to the complete medium. Infected cells were permeabilized by incubating them with 0.025% Triton X-100 (Sigma-Aldrich) for 20 min at the indicated time points. Afterward, cell lysates were serially diluted and plated onto 7H11 agar plates supplemented with 10% OADC. Bacterial colony-forming units (CFUs) were quantified after 3 weeks of incubation. Each sample was plated in triplicate.

For the infection assay, phorbol 12-myristate 13-acetate (PMA, Sigma-Aldrich)-differentiated THP-1 cells were infected with wild-type, B1033, or B1033C BCG strain at an MOI of 10 for 2 h at 37 °C. Then, the cells were washed three times with sterile 1× PBS to remove extracellular bacteria. This time point was considered 0 h; the cells were cultured in a complete medium with 100 μg/mL gentamicin for various durations (12 h, 24 h, 48 h, and 72 h). The cells and supernatants were collected via centrifugation for further analysis.

### 2.4. Sequencing of Mutants

Genomic DNA was extracted using the cetyltrimethylammonium bromide method [16]. The presence of mutations was determined using sequencing amplicons of the transposon–genome junctions as previously described [17]. Upstream and downstream primers were designed following the insertion site and transposon insertion orientation. Sequencing results revealed a potential mutation of *fad*D*33* in the mutant strain B1033. To validate the potential transposon insertion site of B1033 identified through sequencing, we performed PCR using upstream primers derived from transposon sequences and downstream primers derived from sequences 500 to 700 bp downstream of the insertion site (forward—5′-CCTCGTGCTTTACGGTATCGC-3′ and reverse—5′-TTTTCCAGACG-ACGTCCGGG-3′). We used B1033 genomic DNA as the template and ctrl-BCG as the negative control. The PCR conditions were as follows: initial denaturation at 95 °C for 5 min; 30 cycles of denaturation at 95 °C for 30 s, annealing at 60 °C for 1 min, and extension at 72 °C for 1 min; and final extension at 72 °C for 5 min. A positive band of the expected size in the mutant strain and no band in the control group confirmed the accuracy of the sequencing results and the location of the inserted gene.

### 2.5. Construction of the BCG Complement Strain

Full-length *fad*D*33* (GenBank ID: 886855) was amplified from BCG genomic DNA using specific primers (forward: 5′-CGCGGATCCATGAGTGAGCTCGCGGCC-3′ and reverse: 5′-CGCAAGCTTTCAGTCCGCCATCTCCAGGG-3′). The target gene was cloned into the pMV261 vector and electroporated into BCG to generate the recombinant strain B1033C as previously described [18]. To detect the expression of *fad*D*33*, wild-type, B1033, and B1033C BCG strains were cultured at 37 °C in Middlebrook 7H9 broth medium supplemented with 10% OADC, 0.05% Tween 80, 0.2% glycerol, and 150 μg/mL hygromycin until the broth became turbid and the absorbance at 600 nm (OD_600_) reached 0.8–1.0 [19]. The bacteria were then washed and sonicated. Protein samples were separated using 10% sodium dodecyl sulfate-polyacrylamide gel electrophoresis (SDS-PAGE) and transferred to a polyvinylidene difluoride (PVDF) membrane (Millipore, Burlington, MA, USA), which was blocked with 5% bovine serum albumin in Tris-buffered saline containing Tween 20 (TBST). Membranes were incubated overnight at 4 °C with a mouse anti-*fad*D*33* antibody or a mouse polyclonal anti-β-actin (Protech, Wuhan, China, Cat no. 60008-1) diluted 1:1000 in TBST. Protein band intensities were detected using Western Bright ECL (Advansta, San Jose, CA, USA), and β-actin was used as the internal reference. Quantitative analyses of the band intensities were performed using Image J software (Version 1.51j8, National Institutes of Health, Bethesda, MD, USA).

### 2.6. In Vitro Kinetics of Recombinant BCG Strain Growth

BCG strains were cultured in triplicate at 37 °C in 25 mL of 7H9 broth liquid medium supplemented with 10% OADC, 0.05% Tween 80, and 0.2% glycerol until the growth reached the stationary phase. OD_600_ was measured at the indicated time points.

### 2.7. In Vitro Stress Assay

To analyze the impact of acidic or membrane stress on bacterial strains, BCG was incubated in various conditions: (1) a low pH culture condition: pH was adjusted to 4.5 by adding hydrochloric acid (HCl) to 7H9 liquid medium, and the medium was sterilized using an autoclave as previously described [20]; and (2) a membrane stress culture condition: 0.05% sodium dodecyl sulfate was added to 7H9 medium for preparing an oxidative membrane stress model as previously described [20].

Wild-type, B1033, and B1033C cultures were incubated at 37 °C for 12, 24, 36, and 48 h. At each time point, bacterial cells were collected, diluted tenfold, and spotted on 7H11 solid media plates supplemented with antibiotics. The plates containing BCG strains were incubated at 37 °C for 3–4 weeks.

### 2.8. Morphology of BCG and Diameter of Single Colonies

BCG strains were cultured until growth reached the logarithmic phase. The bacterial solution was diluted 10× with Hank’s Balanced Salt Solution (HBSS, Thermo Fisher Scientific Inc., Waltham, MA, USA) buffer to a concentration of 10^−5^ to 10^−8^, and 100 μL of each dilution was spread on a 7H11 plate in triplicate. After a 21-day incubation at 37 °C, the morphology of single colonies was observed and the diameter of three colonies per treatment group was measured using a digital microscope and its associated software (KEYENCE, version VHX-7000, Osaka, Japan).

### 2.9. Cell Wall Permeability Assay

Ethidium bromide (EtBr) accumulation was measured to assess cell wall permeability [21]. BCG was grown in 7H9 medium until it reached the logarithmic growth phase. Then, the cells were washed and resuspended in PBS containing 0.05% Tween 80. The OD_600_ value of the resuspended bacteria was adjusted to 0.8, and 200 μL of the resulting bacterial suspension was incubated with EtBr (2 μg/mL) (Sigma-Aldrich) and plated in triplicate into a 96-well black fluoroplate. Furthermore, the cultures were stained with 1 μg/mL of EtBr for the EtBr assay. The accumulation of EtBr was measured via fluorescence at the indicated time points using a Tecan Infinite 200 Reader with an excitation wavelength of 544 nm and emission wavelength of 590 nm. Data from each well were normalized to those obtained at 0 h. All experiments were performed in triplicate.

### 2.10. Western Blot Assay

Infected cells were collected and lysed in radioimmunoprecipitation assay (RIPA, Sigma-Aldrich) lysis buffer for 30 min on ice. Proteins were separated using 12% SDS-PAGE and electroblotted onto PVDF membranes. Membranes were incubated with 5% free-fat dry milk for 3 h at room temperature and with primary antibodies overnight at 4 °C. The primary antibodies were anti-p38 (Cat no. 8690), anti-phosphorylated-p38 (Cat no. 4511), anti-jun N-terminal kinase (JNK) (Cat no. 9252), anti-phosphorylated JNK (Cat no.4668), anti-extracellular signal-regulated kinase (ERK) (Cat no. 4695), and anti-phosphorylated ERK (Cat no. 4370) (Cell Signaling Technology, Danvers, MA, USA) antibodies, as well as a mouse polyclonal antibody recognizing β-actin (Protech, Wuhan, China, Cat no. 60008-1). After washing with TBS containing 0.05% Tween 80 three times, membranes were incubated with secondary antibodies for 1 h at room temperature. Protein bands were visualized using an ECL kit, and protein expression was analyzed using Image J software (Version 1.51j8, National Institutes of Health, Bethesda, MD, USA).

### 2.11. Enzyme-Linked Immunosorbent Assay (ELISA)

PMA-differentiated THP-1 cells were seeded in 12-well plates at a density of 1 × 10^6^ cells per well and cultured for 12 h. Then, the cells were infected with wild-type, B1033, or B1033C BCG strain at an MOI of 10 for 2 h. For inhibition assay, PMA-differentiated THP-1 cells were pretreated with a p38 inhibitor (SB202190, 10 µM, MedChemExpress, Monmouth Junction, NJ, USA) for 2 h and infected with wild-type, B1033, or B1033C BCG strain. The number of viable bacilli in THP-1 cells was measured 72 h post infection (PI) using a colony-forming unit (CFU) assay. The infected cells were permeabilized with 0.025% Triton X-100 for 20 min at the indicated time points. Afterward, cell lysates were serially diluted, plated onto 7H11 agar plates supplemented with 10% OADC, and incubated at 37 °C for 21 d. The expression levels of IL-1β, IL-6, and TNF-α in cell culture supernatants were determined using an ELISA kit (Neobioscience, Shenzhen, China) following the manufacturer’s instructions. Each sample was plated in triplicate.

### 2.12. RNA Preparation and Reverse Transcription-Quantitative PCR (RT-qPCR)

PMA-differentiated THP-1 cells were infected with bacterial strains at an MOI of 10. At the indicated time points PI, total RNA was extracted, and cDNA was synthesized using HiScript II Q RT SuperMix for qPCR (Vazyme, Nanjing, China). We performed qPCR amplification using Bio-Rad IQ5 and the following PCR conditions: 5 min at 95 °C and 40 cycles of 30 s at 95 °C, 30 s at 60 °C, and 30 s at 72 °C. We used β-actin as the internal control. All primers are listed in Table 1.

### 2.13. Statistical Analysis

All assays and experiments were performed in triplicate. Data are presented as means ± standard errors of the mean (SEMs) of the triplicates. GraphPad Prism software (Version 7.0, Boston, MA, USA) was used for statistical analysis. Statistical significance was determined using Student’s *t*-test for comparison between two groups and analysis of variance (ANOVA) for comparison among more than two groups. A *p* value of <0.05 was considered statistically significant.

## 3. Results

### 3.1. Absence of fadD33 Decreased BCG Invasion and Adhesion Abilities

A BCG transposon library was constructed in our laboratory using MycomarT7 phages containing Himar1 Mariner transposon, which inserts randomly at TA dinucleotides. The library consisted of 3034 mutants, of which more than 200 were screened for their invasion abilities. Among them, B1033 had the lowest invasion ability (*p* < 0.001) (Figure 1a) and a lower adhesion ability than BCG (*p* < 0.01) (Figure 1b). Transposon sequencing and PCR data revealed that the gene mutated in B1033 cells was *fad*D*33* (Appendix A), indicating that the absence of *fad*D*33* in BCG reduced its invasion and adhesion abilities.

### 3.2. fadD33 Decreased the Size of BCG Colonies

To further investigate the function of *fad*D*33* in BCG, we constructed a B1033 complemental strain, B1033C (Appendix A). Upon cultivation on 7H11 agar plates, distinct morphological differences were observed among wild-type BCG, B1033, and B1033C strains. B1033 showed a significantly enlarged colony morphology compared with wild-type BCG and B1033C (*p* < 0.001). However, upon complementation of *fad*D*33*, the colony diameter reverted to a size comparable to that of wild-type BCG (Figure 2). These observations suggest that *fad*D33 may play a role in regulating the colony size of BCG strains.

### 3.3. Absence of fadD33 Decreased the Extracellular and Intracellular Survival under Stress Conditions

During infection, mycobacteria encounter a diverse range of bactericidal stresses, both extracellularly and intracellularly. To evaluate the adaptability of *fadD33* to various stress environments, we determined the survival of B1033 under low pH, membrane stress, and intracellular stress conditions. Under regular extracellular culture conditions, B1033 and BCG showed a similar growth curve (*p* > 0.05) (Figure 3a). However, under low pH and membrane stress conditions, the extracellular survival rate of B1033 was significantly reduced compared with BCG at 24 h and 12, 24, and 36 h, respectively (*p* < 0.001) (Figure 3b,c). Furthermore, B1033 demonstrated significantly reduced intracellular survival compared with wild-type BCG at 72 h PI (*p* < 0.001) (Figure 3d). These findings suggest that the absence of *fad*D*33* slows down the intracellular and extracellular growth of BCG under stress conditions.

### 3.4. Absence of fadD33 Increased Bacterial Cell Wall Permeability

The robust lipophilic cell wall of mycobacteria serves as a vital physical barrier, safeguarding them against adverse intracellular environments [25,26,27]. To ascertain the function of *fad*D33, we conducted cell permeability assays. Significantly, there was a notable increase in EtBr accumulation in the B1033 group compared to the control group, observed as early as 10 min post-EtBr treatment (*p* < 0.001) (Figure 4). This suggests that the absence of *fad*D33 in BCG increased cell wall permeability.

### 3.5. fadD33 Regulated the p38 MAPK and ERK Signaling Pathways

To elucidate the signaling pathway through which *fad*D33 modulates intracellular survival, Western blot analyses of p-p38/p38, p-JNK/JNK, and p-ERK/ERK expression were conducted. Our findings revealed that the ratios of p-p38/p38 and p-ERK/ERK expression in the B1033 infection group were significantly elevated at 24 h PI compared with those in the wild-type and B1033C infection groups (*p* < 0.001), with no significant difference between the wild-type BCG and B1033C infection groups (*p* > 0.05). However, for the JNK pathway, the ratio of p-JNK/JNK in the B1033 infection group was significantly lower than that in the wild-type BCG group at 24 h PI (*p* < 0.001), with an even lower ratio observed in B1033C (*p* < 0.001) (Figure 5). These results suggest that *fad*D33 downregulates the p38 and ERK signaling pathways during mycobacterial infection but not the JNK pathway.

### 3.6. fadD33 Inhibited the Expression of Proinflammatory Cytokines through the p38 MAPK Signaling Pathway

To clarify whether *fad*D*33* regulated the release of proinflammatory cytokines in the host, IL-1β, IL-6, and TNF-α expressions were examined. IL-1β, IL-6, and TNF-α mRNA expressions were significantly induced 12 h after infection with B1033, and when complemented with *fad*D*33*, these mRNA expressions reverted to those observed in the wild-type BCG strain (Figure 6a,c,e). A similar trend was observed for IL-1β and IL-6 protein levels at 12 and 24 h PI, respectively (Figure 6b,d). TNF-α protein expression was not significantly different between the infection groups (Figure 6f). In conclusion, *fad*D*33* inhibited the expression of IL-1β and IL-6 and the transcription of TNF-α but not protein secretion at 12 and 24 h PI.

Then, THP-1 cells were pretreated with the p38 inhibitor SB202190 for 2 h prior to infection. The levels of IL-1β, IL-6, and TNF-α in the culture supernatants were quantified at 72 h PI. Notably, the levels of IL-1β, IL-6, and TNF-α were significantly elevated following infection with B1033 compared with those in the wild-type (*p* < 0.001). In contrast, the B1033C infection group exhibited expression levels comparable to those of wild-type BCG (*p* > 0.05). Upon inhibition of p38, the expression of all three cytokines was significantly reduced (*p* < 0.05), except for IL-6 expression in the B1033C infection group (*p* > 0.05) (Figure 7). These findings suggest that *fad*D33 inhibits the expression of IL-1β, IL-6, and TNF-α via the p38 signaling pathway.

### 3.7. fadD33 Promoted BCG Intracellular Survival through the p38 MAPK Signaling Pathway

THP-1 cells were pretreated with the p38 inhibitor SB202190 for 2 h prior to infection with BCG, B1033, or B1033C. Given that *fad*D33 absence diminished the intracellular survival of BCG, the inhibition of p38 resulted in a significant increase in the intracellular survival of B1033 and B1033C (*p* < 0.001, *p* < 0.05) (Figure 8).

These findings suggest that *fad*D33 enhances the intracellular survival of BCG by modulating the p38 MAPK signaling pathway.

## 4. Discussion

In the present study, we revealed that the absence of *fad*D*33* in BCG inhibits its survival in adverse extracellular and intracellular microenvironments, reduces its invasion and adhesion capabilities, alters its morphology, and increases its cell wall permeability. By constructing a complemental strain, we further confirmed that *fad*D*33* suppresses the expression of proinflammatory cytokines via the p38 MAPK signaling pathway.

### 4.1. fadD33 Enhanced the Ability of Mycobacterium to Counteract Host Cells

Previous studies have identified *fad*D33 as a fatty acyl AMP ligase that is involved in the synthesis of mycobactin and is crucial for maintaining the integrity of the cell wall [14]. Our current findings demonstrated that the absence of *fad*D33 in mycobacteria can diminish their invasion and adhesion capabilities. Given that mycobacteria are intracellular pathogens capable of persistent infection within host cells, a reduction in their adhesion and invasion abilities undoubtedly significantly impacts their infection and persistent infection of host cells.

Upon successful infection of host cells, mycobacteria encounter a wide range of intracellular stresses, including hypoxia, nutrient scarcity, acidic pH, and membrane tension [3,4,5]. Our findings revealed that *fad*D*33* does not affect the growth rate of BCG under regular growth conditions. However, under stress conditions, the absence of *fad*D*33* inhibits the survival of BCG, which could be beneficial for host cells.

Meanwhile, the absence of *fad*D*33* was found to increase the permeability of the mycobacterial cell wall. Meanwhile, the absence of *fad*D33 increased the permeability of the mycobacterial cell wall. The thick, lipophilic cell wall serves as a crucial physical barrier for mycobacteria, protecting them against hostile intracellular environments and rendering them highly impermeable to antimicrobial molecules produced by the host immune system upon infection [25,26,27]. Thus, mycobacteria loss of cell wall integrity can decrease the anti-host cell ability of mycobacterium. This finding is consistent with that of other studies. For instance, it has been reported that the *M. tb* mutant ΔmbtE, which results in impaired mycobactin synthesis, affects cell wall permeability and weakens the growth of intracellular strains [28]. Similarly, the *M. tb* protein, Rv2387, which is involved in fatty acid metabolism, leads to significant changes in the fatty acid composition of the cell wall and increased sensitivity to acidic stress [29]. Another protein, Rv3272, has been shown to reduce cell wall permeability and cause triacylglycerol accumulation on the cell wall, ultimately altering the cell wall lipid distribution and thereby protecting mycobacteria from acidic and oxidative stresses and promoting their survival within cells [30]. Given that the absence of *fad*D*33* also decreased the invasion and adhesion abilities, as well as the survival of mycobacterial under low pH and membrane pressure conditions, we inferred that *fad*D*33* might facilitate mycobacterial resistance to stress and increase virulence by decreasing cell wall permeability. During the interaction of host cells and mycobacterium, *fad*D*33* can increase the anti-host cell ability of mycobacterium.

### 4.2. fadD33 Inhibited Proinflammatory Cytokines through the p38 MAPK Signaling Pathway

Macrophages play a crucial role in exerting antimicrobial control and initiating and maintaining inflammation [31]. The p38 MAPK pathway is a critical regulator of inflammation [32]. Our study revealed that the absence of *fad*D33 significantly elevated the expression of IL-1β, IL-6, and TNF-α in THP-1 cells, which was confirmed by *fad*D33 complementation. *fad*D33 appears to inhibit proinflammatory cytokine production via the p38 MAPK signaling pathway to promote the intracellular survival of BCG.

Inflammatory response to *M. tb* infection is a double-edged sword. While cytokines such as TNF-α and IL-1 are important for protection against infection, excessive or insufficient cytokine production can result in diseases [31]. Mice lacking IL-1β and IL-1α or TNF-α have been reported to be more susceptible to *M. tb*. *fad*D13, one of the 36 homologs of *fad*D in *E. coli*, is crucial for *M. tb* proliferation in macrophages and plays a key role in the production of proinflammatory cytokines during *M. tb* infection [33]. In this study, we found that *fad*D33 can inhibit the secretion of proinflammatory cytokines via the p38 MAPK pathways, thereby promoting the intracellular survival of mycobacterium. This finding is consistent with our earlier data indicating that the absence of *fad*D33 decreases both intracellular and extracellular survival of mycobacterium.

In summary, our study revealed that the absence of *fad*D33 decreased the adhesion and invasion abilities of mycobacteria and increased its cell wall permeability, thereby diminishing its entry and survival in host cells, especially under low pH and high membrane pressure conditions. In addition, *fad*D33 promoted the intracellular survival of BCG in macrophages by inhibiting the release of proinflammatory cytokines via the p38 MAPK signaling pathway (Figure 9).

## 5. Conclusions

The absence of *fad*D*33* decreased the adhesion and invasion of BCG in A549 cells and increased the permeability of the mycobacterial cell wall, thereby inhibiting its survival under low pH and high membrane pressure conditions within the host’s extracellular microenvironment. *fad*D*33* can promote BCG intracellular survival in macrophages by suppressing the release of proinflammatory cytokines, such as IL-1β, IL-6, and TNF-α. This study uncovers a novel survival mechanism employed by *M. tb* to evade host defenses, which can potentially serve as a therapeutic and vaccine target for tuberculosis prevention.

## Figures and Tables

**Figure 1 cells-12-02610-f001:**
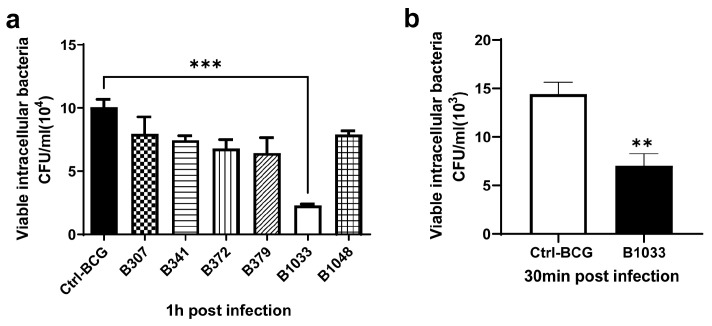
Invasion and adhesion assays of BCG strains. A549 cells were infected with BCG strains at an MOI of 10 for 30 min at 4 °C for adhesion assay and 1 h at 37 °C for invasion assay. The numbers of intracellular BCG were quantified using CFUs. (**a**) Invasion ability of different BCG mutants. (**b**) Adhesion ability of B1033 mutant and BCG. Each sample was plated in triplicate. Data are presented as means ± SEMs. ** *p* < 0.01 and *** *p* < 0.001.

**Figure 2 cells-12-02610-f002:**
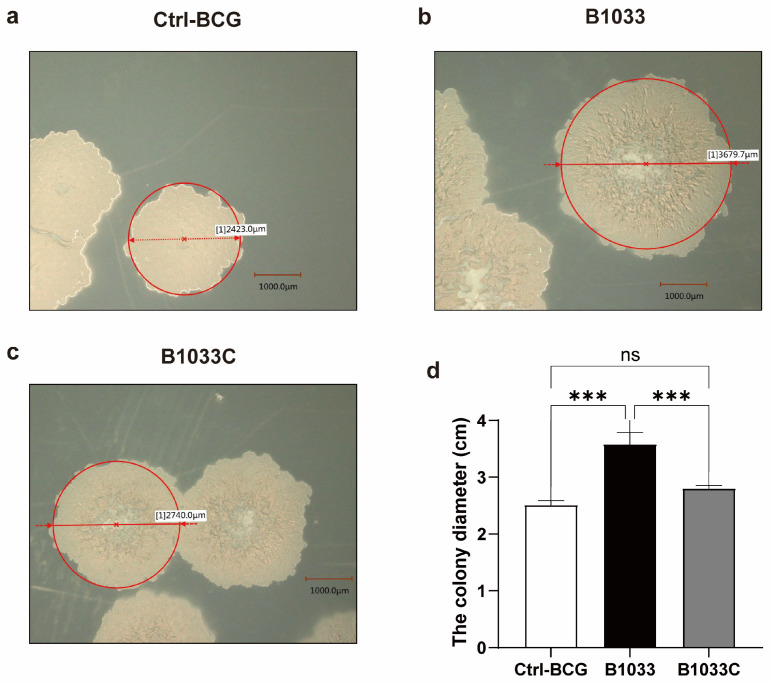
Morphology and diameter of single colonies of wild-type (**a**), B1033 (**b**), and B1033C (**c**) BCG strains. (**d**) Diameter of wild-type colonies (ctrl-BCG), B1033, and B1033C BCG strains. ns *p* > 0.05; *** *p* < 0.001.

**Figure 3 cells-12-02610-f003:**
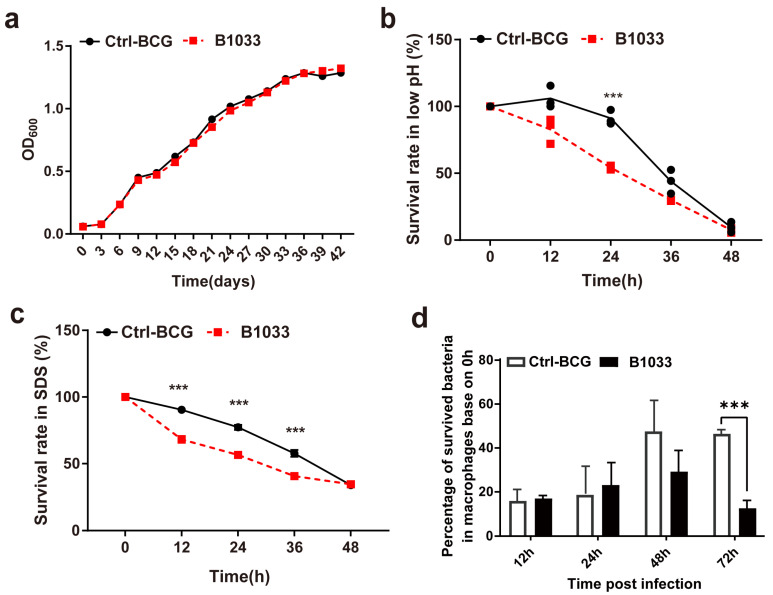
Extracellular and intracellular survival of B1033. Survival rate of B1033 and wild-type BCG (Ctrl-BCG) strains incubated in (**a**) 7H9 medium at 37 °C, (**b**) low pH (pH 4.5) 7H9 liquid medium, (**c**) 7H9 liquid medium containing 0.05% SDS, and (**d**) THP-1 cells. *** *p* < 0.001.

**Figure 4 cells-12-02610-f004:**
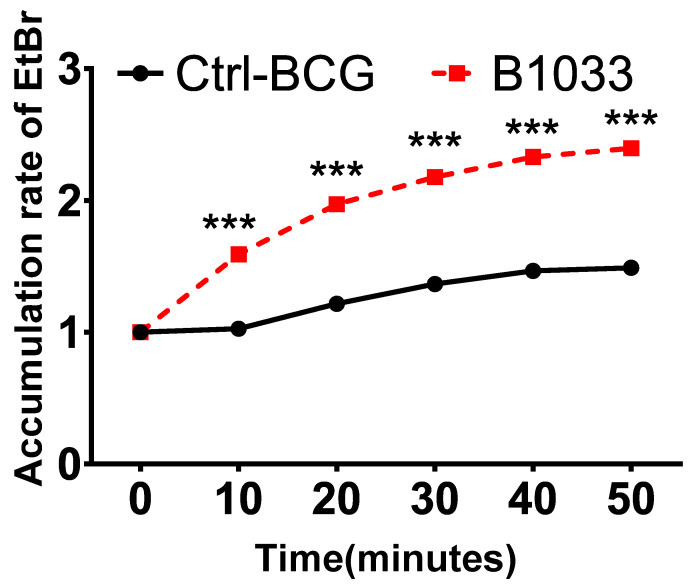
Effect of *FadD33* on cell wall permeability. Wild-type BCG (ctrl-BCG) and B1033 BCG strains were treated with 2 µg/mL EtBr. The relative fluorescence of intracellular EtBr was measured at 10 min intervals. *** *p* < 0.001.

**Figure 5 cells-12-02610-f005:**
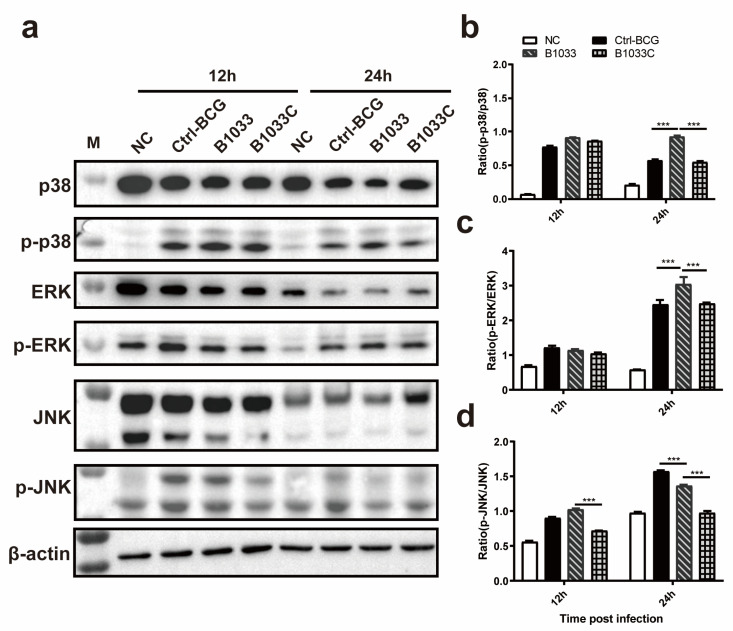
Expression of p38, ERK1/2, and JNK. THP-1 cells were infected with B1033, B1033C, and wild-type BCG (Ctrl-BCG) strains at an MOI of 10. (**a**) Western blot analysis of p38, phosphorylated p38 (p-p38), ERK1/2, phosphorylated ERK1/2 (p-ERK), JNK, and phosphorylated JNK (p-JNK) expression 12 and 24 h PI. (**b–d**) Ratios of the levels of phosphorylated to unphosphorylated protein for p38 (**b**), ERK1/2 (**c**), and JNK (**d**). *** *p* < 0.001.

**Figure 6 cells-12-02610-f006:**
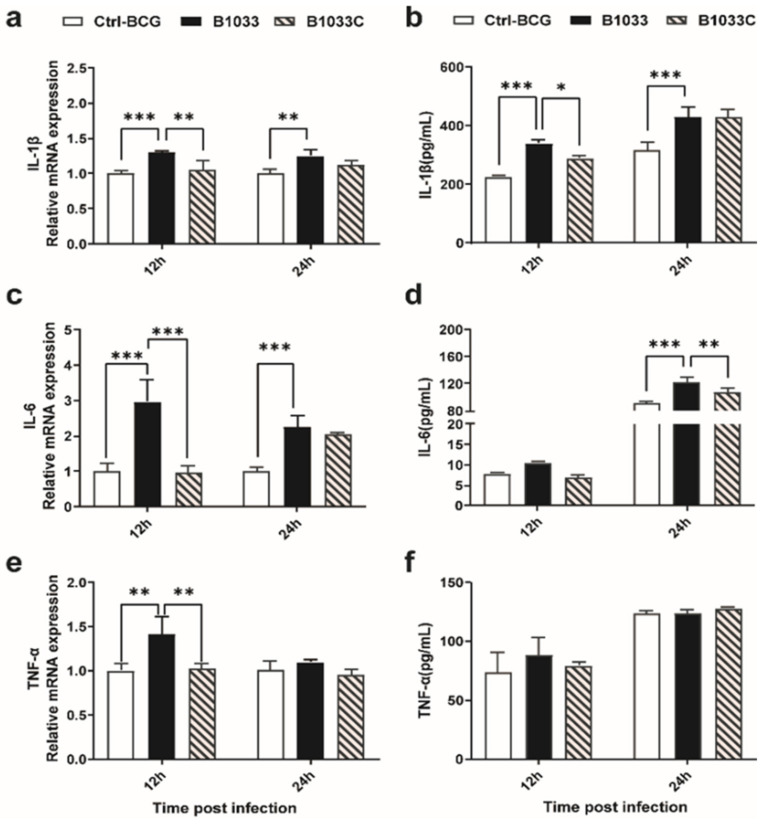
Expression of IL-1β, IL-6, and TNF-α in THP-1 cells. THP-1 cells were infected with wild-type BCG, B1033, and B1033C strains. The mRNA levels of (**a**) IL-1β, (**b**) IL-6, and (**c**) TNF-α were detected using RT-qPCR and normalized to β-actin. Cytokine protein levels of (**d**) IL-1β, (**e**) IL-6, and (**f**) TNF-α were detected using ELISA. * *p* < 0.05; ** *p* < 0.01; *** *p* < 0.001.

**Figure 7 cells-12-02610-f007:**
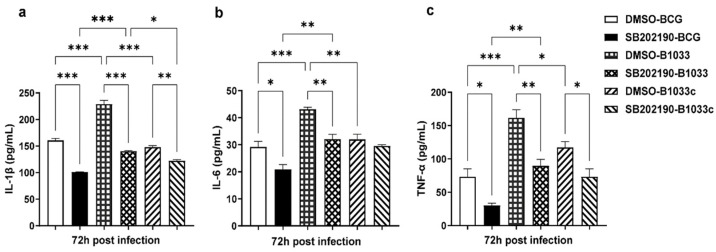
IL-1β, IL-6, and TNF-α expression in THP-1 cells after p38 inhibition. PMA-differentiated THP-1 macrophages were treated with a p38 inhibitor (SB202190, 10 µM) or dimethyl sulfoxide (DMSO) for 2 h and then infected with wild-type (BCG), B1033, and B1033C strains at an MOI of 10. IL-1β (**a**), IL-6 (**b**), and TNF-α (**c**) expressions were detected via ELISA 72 h PI. * *p* < 0.05, ** *p* < 0.01, *** *p* < 0.001.

**Figure 8 cells-12-02610-f008:**
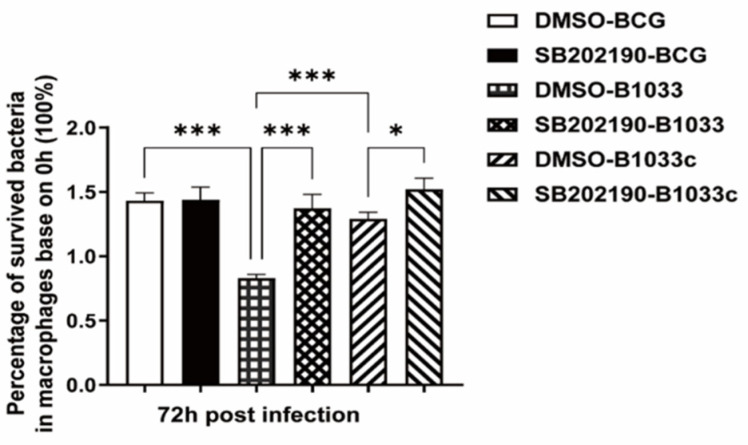
Intracellular survival of BCG after p38 inhibition. PMA-differentiated THP-1 macrophages were treated with a p38 inhibitor (SB202190, 10 µM) or DMSO for 2 h and then infected with wild-type (BCG), B1033, or B1033C strains at an MOI of 10. Survival, expressed as CFU/mL, was assessed at 0 and 72 h PI. The intracellular survival of BCG after 72 h is expressed as a percentage of that at 0 h. * *p* < 0.05, *** *p* < 0.001.

**Figure 9 cells-12-02610-f009:**
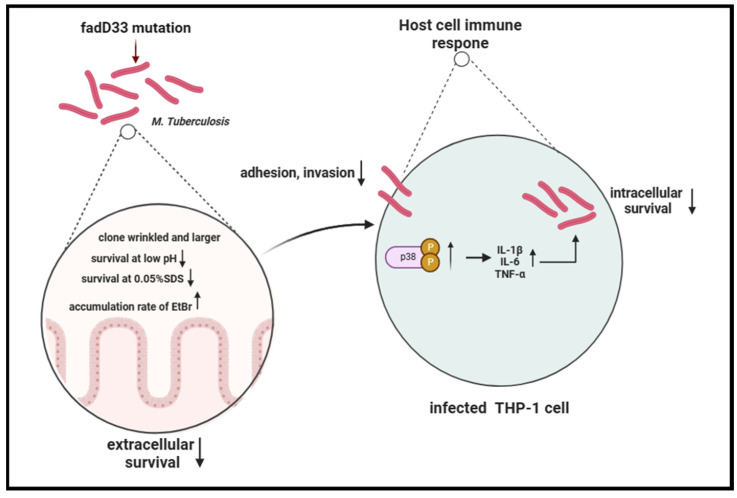
Schema of the function of *fad*D33 during *M. tb* interaction with the host.

**Table 1 cells-12-02610-t001:** Primers used for qPCR.

Primer Name	Sequence (5′-3′)	Products (bp)	Reference
hsa-β-actin	F: CATGTACGTTGCTATCCAGGC R: CTCCTTAATGTCACGCACGAT	250	[22]
hsa-IL-1β	F: GTGGCAATGAGGATGACTTGTTC R: GGTGGTCGGAGATTCGTAGCT	120	[23]
hsa-IL-6	F: ACTCACCTCTTCAGAACGAA R: CCATCTTTGGAAGGTTCAGG	149	[24]
hsa-TNF-α	F: GGAGAAGGGTGACCGACTCA R: CTGCCCAGACTCGGCAA	70	[23]

## Data Availability

The datasets generated in this study are available upon request from the corresponding authors.

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
