# Peer review of "Mycobacterium tuberculosis* Fatty Acyl-CoA Synthetase *fad*D33 Promotes *Bacillus Calmette–Guérin* Survival in Hostile Extracellular and Intracellular Microenvironments in the Host"

_cells, 2023, doi:10.3390/cells12222610_

Round 1

Reviewer 1 Report

Comments and Suggestions for Authors

General Comments:

The manuscript by Zhu et al describes the phenotypes of a BCG strain with a himar insertion into FadD33.   The writing in general is adequate, although some passages could be written to be easier to understand. Also, the authors seem to know very little about tuberculosis. The study is interesting because it shows how a gene involved in metabolism of lipids, presumably on the cell envelop, affects permeability, survival in stress conditions and in macrophages, and the response of the macrophages to infection.  There are a few problems with overinterpretation of the cytokine data, but this can be corrected.  The main aspect that is lacking is a biochemical characterization of the lipids that are missing or altered by the insertion, at least with thin layer chromatography.  Mass spectrometry would be even better, although it can be difficult to find a lab performing MS on mycobacterial lipids. While TLC may not show any obvious differences in the mutant strain, the technique is not hard to perform and if there are no obvious differences this could be stated.

Specific Comments:

Bermudez appears to be misspelled.

Abstract:

One report that the fadD33 affects bacterial proliferation in the liver probably does not warrant mention in the abstract.

Line 22 What are the various kinases and cytokines? Be specific.

Line 24, and elsewhere: It would be better to describe increased permeability (and other phenotypes) in the mutant, rather than inferring that this is what FadD33 does. The actual role of FadD33 is not defined, just the phenotypes of its absence.

Line 26 again, FadD33’s absence decreases survival.

Introduction:

Line 35. TB is not a zoonosis. “consumption” is odd here.

Line 36 TB is again the leading single infectious killer. And there are over 10 Million cases a year.  1.4 million deaths include HIV co-infected people.

Lines 40 – 41 are repetitive “obligatory…obligatory

Line 46-47 and acidification of the phagolysosome… this sentence should be rewritten.

Line 50 contributes

Line 55. You found that an fadD33 mutant had reduced ability etc.

Line 60 is overstating the findings.  BCG without FadD33 survive in culture and within macrophages, albeit less well, so it is not a good target.   It’s not even clear what it does.

Methods:

Line 101 What are the insertion characteristics of the fadD33 gene?

Lines 140 What is meant by “The cultures were further stained with EtBr?”

Line 163 More details is needed on colony counts for intra-macrophage survival. How were the cells lysed, for instance?  Were they diluted?

Results:

Figure 1.  What are the other mutants in a. ? The expt should also be done with the complemented B1033C. “Post infection” would be better. 

Line 208. Isn’t it the absence of FadD33 that causes increased colony size and wrinkles?

Figure 2b.  Could there be a better, larger photo of one or two colonies that show the wrinkles better?

Line 223 “membrane stress environments”. The only stresses shown are acidic media and SDS.  The Y axis in Fig 3C – how can survival be > 100%, but the Y axis goes to 150%  The complemented mutant should be included in the studies shown in Fig 3, and their description in the text should be more detained and precise. 

If figure 1 shows decreased phagocytosis of the mutant at 1 hour, how can Figure 3d show an equal number of intracellular bacteria at 12 and 24 hours?

Figure 4. the complemented mutant is missing, again. Also, the description in the methods of how the experiment was performed could be clearer.

Figure 5 Why is JNK significantly lower in the B1033 mutant and even lower in the complemented B1033C?  This is unexpected and hard to explain.  It contradicts the narrative that the differences are due to the absence of FadD33.  The complemented strain in b is B1033-C, but elsewhere it is B1033C (no hyphen).

Figure 6 has similar problems: at 24 hours neither Il-2B mRNA nor Il-1B protein levels in the complemented B1033C  return to the control level.  Same for IL-6 mRNA at 24 hours, and TNF shows a difference only in mRNA at 12 hours.  However, Line 262-3 states that the FadD33 inhibited the expression of all 3 cytokines, which seems to overstate the results, and is different from the results in 7c., where there is a large increase in TNF expression at 72 hours.  Figure 6 has 12 and 24 hours, so figures 6 and 7  are not completely comparable, which introduces further doubt about the consistency of the results. TNF protein goes down in all 3 strains after p38 inhibition, so not clear this is related to the absence of FadD33.  IL-6 isn’t lower in the complemented B1033C strain with p38 inhibitor.

Figure 8.  Isn’t it possible that FadD33 doesn’t modulate the p38 signalling pathway, but just protects against the p38 induced effects?   Why isn’t the complemented strain like the BCG control?

The authors try to suggest that the data is cleaner, clearer and more consistent than they really are.  The problem is mainly that the complemented strain doesn’t always behave like the control BCG.

Discussion

Generally the first paragraph of the discussion reviews the basic findings of the study.  Here the first two paragraphs are  general and don’t state all the salient results.

The next paragraph only deals with the findings in a roundabout way, and talks about adhesion when what is shown is intracellular uptake at 1 hour.  Is this differences in adhesion or phagocytosis?  Decreased phagocytosis would be beneficial to BCG survival.

Line 343 overstates the results with TNFa

Again, it is better to state that the absence of FadD33 caused the phenotypes, not directly that FadD33 is promotes BCG survival.  It could play an important role in maintaining the integrity of the cell envelop, but not clear how.  This is subtly different from promoting survival. How p38 is involved is not clear, and it is not shown that it, by itself inhibits release of cytokines, but it might just be the integrity of the cell envelope.

It would be better to start with the results and then compare to other similar results, rather than the other way around.

Figure 9 is not clearly needed, and the font within the circles is so small as to be difficult to read.  The results shown in Fig 9 should be stated at the beginning of the discussion, and then this figure wouldn’t be necessary.

From the changed colony morphology, it is likely that some surface exposed lipids are changed, but this is not mentioned in the discussion.

Conclusions:

The problem is that it is not known how FadD33 works, or what it does.  It contributes to these phenotypes, but perhaps as part of the cell envelop structure.  It doesn’t appear to be through mycobactins.

Overall, it would be better to state that FadD33 plays a role in membrane or cell envelope integrity and reduces some cytokines, perhaps by masking antigens  Decreased macrophage uptake is balanced by reduced intramacrophage survival.  Neither the data nor the interpretation are as simple as presented. Does alteration of any fatty-acyl AMP ligase cause similar phenotypes? This could be reviewed briefly. Can you speculate on the function of the FadD33?

Reviewer 2 Report

Comments and Suggestions for Authors

In the study under review the authors analyze the molecular mechanisms underlying the ability of fadD33 gene to regulate the virulence of Mycobacterium tuberculosis.  

Major cocerns:

1. There is no information on bacterial adhesion assay in Materials&Methods section.

2. Description to Figure 1c lacks information on how adhesion ability of BCG and mutants was quantified.  

3. There is no information in Materials&Methods section on how the size of colonies and numbers of wrinkles were measured.

Minor concerns:

1. I do not understand why TB is reffered to as "consumption zoonosis" (line 35). The most common route of human TB infection is aerosol, from person to person.

2. Line 38. According to WHO report 2022, there were 10.6 million new TB cases in 2021, only 0.7 million of which in HIV-positive persons (https://www.who.int/teams/global-tuberculosis-programme/tb-reports/global-tuberculosis-report-2022/tb-disease-burden/2-1-tb-incidence). 

3. Fig.1b repeats informamtion from Fig.1a, and thus is not necessary.

Comments on the Quality of English Language

I would suggest editing of the paper by some native English speaker.

Reviewer 3 Report

Comments and Suggestions for Authors

Tuberculosis is a global health problem. It causes a large number of deaths, disability and it is difficult to treat in the case of multidrug resistance. The search for new therapeutic targets is no doubt. In the paper under review the authors investigate the effect of fatty acyl-CoA synthetase fadD33 to mycobacteria viability in BCG infection model.

Here are some minor comments:

1.      Figures 1a and 1b show the same data. Please explain the difference and the need for fig 1b

2.      Please indicate how many colonies were measured to plot a graph 2d

3.      Figure 3d. What kind of THP-1 cells did you use for that? (suspension monocytic culture or PMA-modified). Please clarify.

Round 2

Reviewer 1 Report

Comments and Suggestions for Authors

Previous review comments have been only partially addressed.  The corrections appear to have been made in a very hasty manner that left many grammatical lapses that require careful copy editing. The language was not a serious problem in the original version.  TB is still not a zoonosis.  Also, in  correcting the manuscriot to make the phenotypes the result of the mutation, rather than showing the specific role of the FadD33, the writing now frequently confuses whether the mutation increases or decreases the phenotype.  For example, in the abstract ln 25, the mutation decreased the permeability.  Then, in lines 275-6,  "the absence of fadD33 in 275 BCG increased cell wall permeability", which is correct and repeated in lines 362 and 366.  However, in line 426, "fadD33 mutation decreases the permeability".

The discussion is still too long and confusing.  The introduction contains now contains a very brief summary of the roles of other FadD proteins, but again, it seems to have been hastily assembled.

Comments on the Quality of English Language

The revised MS has several passages that are not grammatically correct and needs good copy editing.

Author Response

Comments and Suggestions for Authors

Previous review comments have been only partially addressed.  The corrections appear to have been made in a very hasty manner that left many grammatical lapses that require careful copy editing. The language was not a serious problem in the original version.  TB is still not a zoonosis.  Also, in  correcting the manuscriot to make the phenotypes the result of the mutation, rather than showing the specific role of the FadD33, the writing now frequently confuses whether the mutation increases or decreases the phenotype.  For example, in the abstract ln 25, the mutation decreased the permeability.  Then, in lines 275-6,  "the absence of fadD33 in 275 BCG increased cell wall permeability", which is correct and repeated in lines 362 and 366.  However, in line 426, "fadD33 mutation decreases the permeability".

The discussion is still too long and confusing.  The introduction contains now contains a very brief summary of the roles of other FadD proteins, but again, it seems to have been hastily assembled.

Comments on the Quality of English Language

The revised MS has several passages that are not grammatically correct and needs good copy editing.

Response:

Thank you for your kind comments. We carefully revised the whole manuscript as your suggested last time again, including the language, the statement of the abstract, introduction, results and discussion. According to the WHO, OIE (WOAH), FAO and UN, we insist the TB is a zoonosis, as we responded it last time. The relative materials are as follows. In order to avoid the confusion of whether the mutation increases or decreases the phenotype, we changed the “fadD33 mutation” to “the absence of fadD33”, for the data that we confirmed with both fadD33 mutation and complemental strain, we used the expression of “fadD33” only. For the discussion, we rearranged it and used a much clearly statement. For details, could you please see the manuscript. Thank you again!

References

  • Olea-Popelka F, Muwonge A, Perera A, Dean AS, Mumford E, Erlacher-Vindel E, Forcella S, Silk BJ, Ditiu L, El Idrissi A, Raviglione M, Cosivi O, LoBue P, Fujiwara PI. Zoonotic tuberculosis in human beings caused by Mycobacterium bovis-a call for action. Lancet Infect Dis. 2017 Jan;17(1):e21-e25. doi: 10.1016/S1473-3099(16)30139-6. Epub 2016 Sep 30. PMID: 27697390.
  • World Health Organization. Roadmap for zoonotic tuberculosis[J]. 2017.
  • Cosivi O, Grange J M, Daborn C J, et al. Zoonotic tuberculosis due to Mycobacterium bovis in developing countries[J]. Emerging infectious diseases, 1998, 4(1): 59.
  • Müller B, Dürr S, Alonso S, et al. Zoonotic Mycobacterium bovis–induced tuberculosis in humans[J]. Emerging infectious diseases, 2013, 19(6): 899.
  • De la Rua-Domenech R. Human Mycobacterium bovis infection in the United Kingdom: Incidence, risks, control measures and review of the zoonotic aspects of bovine tuberculosis[J]. Tuberculosis, 2006, 86(2): 77-109.
  • Kock R, Michel A L, Yeboah-Manu D, et al. Zoonotic tuberculosis–the changing landscape[J]. International Journal of Infectious Diseases, 2021, 113: S68-S72.

Round 3

Reviewer 1 Report

Comments and Suggestions for Authors

The revised version of the manuscript on FadD33 by Zhu et al is improved in many ways but still has problems. 

There are 3 basic problems:

1)    The MS needs good copy editing.  Does the journal perform this editing?

2)    The introduction seems to have been written in haste, and again, Tuberculosis is not a “old chronic consumption zoonosis”.  

“In 2021, there are about 6.4 million new cases of tuberculosis and 1.4 million tuber-33 culosis-related deaths among human immunodeficiency virus-negative people globally”  There were more than 10 million cases of TB in 2021, and don’t the 1.4 million TB deaths include people co-infected with HIV?

3)    The complemented version of mutant B1033 does not complement all the phenotypes in the mutant:

a.     It is not included in Figures 1, 3 or 4.

b.    It doesn’t complement IL-1B protein in Figure 6 and behaves strangely in Figure 8. This should be addressed in the text.

Minor:

Gentamycin is not mentioned in the methods.

Overall, the MS shows that a lot of experimental work was done and the results are worth reporting, but more effort is needed to improve the presentation of the results.  

Comments on the Quality of English Language

Copy editing is needed.

Author Response

Dear Review,

Thank you very much for sending us the comments. We appreciate the Reviewers’ warm work earnestly, and hope that the correction will meet with approval.

Once again, thank you very much for your comments and suggestions.

With best regards

1) The MS needs good copy editing. Does the journal perform this editing?

Response1: Thank you for your comments.

2) The introduction seems to have been written in haste, and again, Tuberculosis is not a “old chronic consumption zoonosis”. “In 2021, there are about 6.4 million new cases of tuberculosis and 1.4 million tuberculosis-related deaths among human immunodeficiency virus-negative people globally” There were more than 10 million cases of TB in 2021, and don’t the 1.4 million TB deaths include people co-infected with HIV?

Response 2: Thank you for your comments. In the Global Tuberculosis Report 2022, PAGE2, Box 1, paragraph 2 and 4, we got the cited information: there are about 6.4 million new cases of tuberculosis and 1.4 million tuberculosis-related deaths among human immunodeficiency virus-negative people globally.

3) The complemented version of mutant B1033 does not complement all the phenotypes in the mutant:

  1. It is not included in Figures 1, 3 or 4.
  2. It doesn’t complement IL-1B protein in Figure 6 and behaves strangely in Figure 8. This should be addressed in the text.

Response 3: Thank you for your kind comments. For comment a), Fig.1 presented the invasion and adhesion assay of B1033, this was just used for screening the mutants and double check the screening results using adhesion assay, so that’s why we didn’t use complement strain here. For Fig.3, this was also a kind of screen assay about the survival, what we focus was intracellular survival, so we added the complemental strain in the detection of intracellular survival assay (Fig.8) based on the results of Fig.3. For Fig.4, we thought this was not a key phenotype in the current research. However, in further study, we will focus on the cell wall permeability of B1033, and we will conduct the complement strain researches.

For comment b), there were two possibilities, for one, Fig.6 we detected the IL-1β expression at 24 h PI and Fig. 7 was at 72 h PI according to the Fig.3d’ data. Although we did not detect a significant recovery of IL-1β at 24 h PI, it was possible that the cytokine expression would be induced more by THP-1 cells at 72 h PI. The second explanation is that the cytokine expression changed at different time points, which is a common phenomenon. The following papers showed similar trends.

(1)   Kumar M, Sahu S K, Kumar R, et al. MicroRNA let-7 modulates the immune response to Mycobacterium tuberculosis infection via control of A20, an inhibitor of the NF-κB pathway[J]. Cell host & microbe, 2015, 17(3): 345-356.

(2)   Zhang X, Li S, Luo Y, Chen Y, Cheng S, Zhang G, Hu C, Chen H, Guo A: Mycobacterium bovis and BCG induce different patterns of cytokine and chemokine production in dendritic cells and differentiation patterns in CD4+ T cells. Microbiology (Reading) 2013, 159(Pt 2):366-379.

(3)   Zhu T, Liu H, Su L, Xiong X, Wang J, Xiao Y, Zhu Y, Peng Y, Dawood A, Hu C et al: MicroRNA-18b-5p Downregulation Favors Mycobacterium tuberculosis Clearance in Macrophages via HIF-1alpha by Promoting an Inflammatory Response. ACS Infect Dis 2021, 7(4):800-810.

4) Gentamycin is not mentioned in the methods.

Response 4: Thank you for your comments. The gentamycin was mentioned in Line 95 in the methods in the latest version, and to make it clear, we added “For infection assay, Phorbol 12-myristate 13-acetate (PMA)-differentiated THP-1 cells were infected with strains at an MOI of 10 for 2 h at 37°C. Then, cells were washed three times with sterilized 1×PBS to remove extracellular bacteria. By taking this time point as 0 h, the cells continued to be cultured in complete medium with 100 μg/mL gentamicin for various times (12 h, 24 h, 48 h, and 72 h). The cells and supernatants were collected by centrifugation for further analysis.” in Line 101 at M & M 2.3.